# The Continuing Question of Adjuvant Therapy in Clear Cell Renal Cell Carcinoma

**DOI:** 10.3390/cancers14246018

**Published:** 2022-12-07

**Authors:** Stephanie A. Berg, Bradley A. McGregor

**Affiliations:** Dana-Farber Cancer Institute, Brigham and Women’s Hospital, Harvard Medical School, Boston, MA 02115, USA

**Keywords:** adjuvant therapy, renal cell carcinoma, immune checkpoint inhibitors, tyrosine kinase inhibitors

## Abstract

**Simple Summary:**

Treatment options after radical nephrectomy for clear cell renal cell carcinoma (ccRCC) have been studied extensively in large randomized clinical trials. Currently, two therapies are approved for patients to receive for one year: pembrolizumab or sunitinib. Newer advances are being developed to help select patients, since approved therapies can cause toxicity. The aim of our review is to discuss past and recent clinical trials that led to the current approvals and upcoming methods for ideal patient selection.

**Abstract:**

Treatment advances in kidney cancer continually evolve. The focus of treatment options continues with oral vascular endothelial growth factor receptor (VEGFR) tyrosine kinase inhibitors (TKIs) or intravenous immune checkpoint inhibitors (ICIs). Multiple trials exploring the role of adjuvant treatment after cytoreductive nephrectomy in high-risk clear cell renal cell carcinoma are currently ongoing. The discovery of biomarkers may help determine which patients benefit from these treatments, but these are not yet available outside clinical studies. Trials with combination therapies are also ongoing, especially using novel therapies with new mechanisms of action, and will hopefully provide more clues on proper patient and therapy selection in the adjuvant setting.

## 1. Introduction

Renal cell carcinoma (RCC) is the most common form of kidney cancer with clear cell RCC (ccRCC) being the most common histologic subtype [1]. Five-year survival rates for localized, early-stage disease approach 93% but drop to 73% for those with locally advanced (lymph node involvement) disease [2]. Once metastatic, median progression-free survival (PFS) for metastatic RCC (mRCC) was only 10–13 months in the pre-immune checkpoint blockade, but due to newer combination therapies with immune checkpoint inhibitors (ICIs) and vascular endothelial growth factor receptor (VEGFR) tyrosine kinase inhibitors (TKIs) or ICIs combinations, median PFS has improved [3]. If a patient with metastatic disease can undergo surgery with curative intent to remove all metastatic sites shortly after cytoreductive radical nephrectomy (CRN), recurrence rates can still approach 100% [4]. To extend survival rates of locally advanced disease at time of diagnosis, adjuvant therapy after radical nephrectomy (RN) has been studied in various populations with different therapeutic agents over the past decade. To date, only one trial incorporating VEGFR TKIs has demonstrated a disease-free survival (DFS) benefit, but without an overall survival (OS) benefit [5]. Adjuvant treatment using a mammalian target of a rapamycin (mTOR) inhibitor, everolimus, demonstrated a numerical improvement in recurrence-free survival (RFS) in patients with RCC after CRN, but it failed to meet the pre-specified statistical threshold for significance [6]. The role of immune checkpoint inhibitors (ICIs) is evolving; therapy with adjuvant pembrolizumab showed an improvement in DFS and a trend toward OS, while trials of other ICIs have thus far been disappointing [7,8,9,10].

## 2. The Foundation for Adjuvant Therapy in RCC

The goal of any adjuvant therapy is to reduce relapse and prolong survival following nephrectomy. This review will cover historical trials in the cytokine era, along with contemporary trials combining VEGFR TKIs and immuno-oncology agents. Summaries will be provided along with comparison tables and a discussion of advantages and disadvantages for each therapy.

### Historical RCC Trials: Cytokine Era

RCC is an immunogenically driven tumor and, prior to the era of ICIs and targeted therapy, two cytokines were tested in the adjuvant setting after RN. A phase III trial conducted by the Cytokine Working Group studied one course of bolus high-dose IL-2 vs. observation in high-risk ccRCC patients (T3b-c, T4, N1–3, or M1 disease resected to no evidence of disease) after CRN. Sixty-nine patients were enrolled, but the trial was terminated early since an interim analysis determined that the 30% improvement in 2-year DFS (primary endpoint) could not be achieved despite full accrual [11]. Furthermore, a single center pilot study explored a 6-month course of subcutaneous IL-2 in the same high-risk ccRCC population; no improvement in DFS or OS was demonstrated [12].

Trials with interferon-α (INF-α) in the adjuvant setting have had similar outcomes. A phase III clinical trial led by the Eastern Cooperative Oncology Group (ECOG) investigated INF-α vs. observation in pT3–4a and/or node-positive RCC patients after CRN. Follow-up at 10.1 years demonstrated no improvement in median OS of INF-α (hazard ratio (HR), 1.35 (95% confidence interval (CI) 0.98–1.86, *p* = 0.07)). [13] An Italian group study explored the cytokine combination of low-dose IL-2 and INF-α vs. observation in RCC patients after CRN with histological RCC subtypes of at least 2.5 cm in size [14]. At a median follow-up of 52 months, RFS and OS were similar (HR, 0.84 (95% CI, 0.54–1.31; *p* = 0.44); HR, 1.07 (95% CI, 0.64–1.79; *p* = 0.79)) [14]. Lastly, a phase III trial led by the European Organization for Research and Treatment of Cancer (Genito-Urinary Cancers Group)/National Cancer Research Institute investigated adjuvant 5-flurouracil, INF-α, and IL-2 vs. observation in high-risk RCC patients after CRN [15]. This treatment was associated with significant toxicity with no improvement in OS at 5 years (63% vs. 70% with treatment (HR, 0.87 (95% CI 0.61–1.23, *p* = 0.428))). Currently, cytokines do not have a role in adjuvant therapy for ccRCC after RN.

## 3. VEGFR TKI Adjuvant Clinical Trials

Loss of the von Hippel Lindau (VHL) tumor suppressor causes accumulation of the hypoxia inducible factor (HIF) complex and subsequent dysregulation of angiogenesis. It has been discovered that biallelic mutations of the *VHL* gene (frequently point mutations) are nearly universal in sporadic ccRCC [16]. VEGFR TKIs that target downstream angiogenic signaling have proven effective in multiple clinical trials for metastatic RCC. There are currently seven approved oral VEGFR TKIs for metastatic ccRCC: axitinib, cabozantinib, lenvatinib, pazopanib, sorafenib, sunitinib, and tivozanib. Bevacizumab, an intravenously delivered anti-VEGF antibody, is also approved as monotherapy [17]. Given their activity in the metastatic setting, multiple trials of adjuvant VEGFR TKIs after RN have been conducted; six large phase III trials are discussed below and summarized in Table 1.

### 3.1. Sunitinib or Sorafenib

The ASSURE (adjuvant sunitinib or sorafenib for unfavorable renal cell carcinoma) phase 3 trial led by the ECOG was the first to use VEGFR TKIs in the adjuvant RCC setting after RN [18]. This trial randomized 1943 patients (between 2006–2010) with pathological-grade pT1b or greater RCC (including variant histologies) following definitive surgical management to receive sunitinib 50 mg daily for 4 weeks on and 2 weeks off, oral sorafenib 400 mg twice daily, or placebo (dosed either continuously or 4 weeks on and 2 weeks off) for one year. The trial suffered from high treatment discontinuation rates ascribable to toxicity (44% from sunitinib and 45% from sorafenib) and therefore, after 1323 patients had enrolled, the starting dose of sunitinib and sorafenib were reduced to 37.5 mg daily for 4 weeks on and 2 weeks off and 400 mg daily, respectively, before an additional 620 patients enrolled. With a median follow-up of 5.8 years, there was no improvement in DFS vs. placebo with either sunitinib or sorafenib; median DFS was 70 months for sunitinib (HR vs. placebo was 1.02 (97.5% CI 0.85–1.23, *p* = 0.8038)), 73.4 months for sorafenib (HR vs. placebo was 0.97 (97.5% CI 0.80–1.17, *p* = 0.7184)), and 79.6 months for placebo [18]. The most common adverse events (AEs) observed in the treatment arms were hypertension (17% patients on sunitinib and 16% patients on sorafenib) and hand–foot syndrome (15% patients on sunitinib and 33% patients on sorafenib).

These results stand in contrast to S-TRAC (sunitinib treatment of renal adjuvant cancer), an international phase III trial that investigated sunitinib vs. placebo in locally advanced ccRCC after surgical management (ccRCC: >pT3 or regional lymph node metastases or both) [5]. Between 2007–2011, 615 patients were randomized to sunitinib or placebo at 50 mg daily on a 4-weeks-on, 2-weeks-off schedule; dose reductions to 37.5 mg daily were allowed. The primary objective was DFS per blinded independent review. At time of initial publication, adjuvant sunitinib improved DFS vs. placebo (HR, 0.76 (95% CI 0.59–0.98, *p* = 0.03)) and median duration of follow-up was 5.4 years in both groups. However, by investigator review, the difference in DFS between sunitinib vs. placebo was not significant (HR, 0.81 (95% CI 0.64–1.02; *p* = 0.08)). On final analysis, OS was not reached in either group (HR, 0.92 (95% CI 0.66–1.28; *p* = 0.6)) with 22% and 24% of patients dying in the sunitinib and placebo arms, respectively [19]. In the sunitinib group, 60.5% of patients had grade 3 or higher treatment-related AEs with palmar–plantar erythrodysesthesia (16%) and hypertension (7.8%) as the most common [5]. In 2017, the United States (US) Food and Drug Administration (FDA) approved sunitinib for the adjuvant treatment of adult patients with ccRCC following RN [20]. Given the contrasting results, a subset analysis of ASSURE explored the high-risk ccRCC population: ≥pT3 or lymph node positive (N+) disease (n = 1069 patients) [21]. In this analysis, five-year DFS rates were 47.7%, 49.9%, and 50% for sunitinib, sorafenib, and placebo, respectively. Similarly, there was no improvement in 5-year OS: 75.2% for sunitinib (HR vs. placebo 1.06 (97.5% CI 0.78–1.45, *p* = 0.66)), 80.2% for sorafenib (HR vs. placebo 0.80 (97.5% CI 0.58–1.11, *p* = 0.12)), and 76.5% for placebo. Similarly, there was no difference in DFS associated with quartile of average dose per cycle (log rank *p*  =  0.38 and 0.79 for sunitinib and sorafenib, respectively) [21].

Unlike the US FDA, the European Medicines Agency did not approve sunitinib as adjuvant treatment in ccRCC patients after RN, thus active surveillance remained the main option for European patients. The SORCE international phase III intergroup (Europe/Australia/New Zealand) trial evaluated sorafenib vs. placebo in patients with RCC (any histology) at intermediate or high risk (per Leibovich risk model) of relapse after nephrectomy [22,23]. The trial randomized 1711 patients between 2007–2013 in a 2:3:3 fashion to 3 years of placebo, one year of sorafenib followed by 2 years of placebo, or 3 years of sorafenib. Initial starting dose of sorafenib was 400 mg twice daily but, as in ASSURE, this was amended two years into trial enrollment to 400 mg daily because of higher-than-expected discontinuation rates [23]. No improvement in DFS was obtained with 3 or 1 year of sorafenib vs. placebo (HR, 1.01 (95% CI 0.82–1.23, *p* = 0.0946); HR, 0.94 (95% CI 0.77–1.14, *p* = 0.509)). Similarly, there was no improvement in 10-year OS rates: 69% for placebo, 70% for 3 years of sorafenib, and 69% for 1 year of sorafenib. In a pre-planned DFS Leibovich high-risk RCC subgroup analysis, no difference in DFS was seen for the 1- or 3-year sorafenib arms. AEs greater than grade 3 occurred in 63.9% of patients taking 3 years of sorafenib, 58.6% of patients taking 1 year of sorafenib, and 29.2% of patients in the placebo group. The most common reported AEs were rash (placebo 30%, 1-year sorafenib 70%, 3-year sorafenib 71%), hand–foot syndrome (placebo 32%, 1-year sorafenib 79%, 3-year sorafenib 77%), diarrhea (placebo 32%, 1-year sorafenib 61%, 3-year sorafenib 64%), hypertension (placebo 48%, 1-year sorafenib 60%, 3-year sorafenib 64%), alopecia (placebo 12%, 1-year sorafenib 54%, 3-year sorafenib 49%), and fatigue (placebo 60%, 1-year sorafenib 74%, 3-year sorafenib 74%) [23].

### 3.2. Pazopanib

The PROTECT (pazopanib as adjuvant therapy in localized/locally advanced RCC after nephrectomy) international phase III trial evaluated pazopanib vs. placebo for patients with clear-cell predominant or ccRCC at high risk for relapse after CRN [24]. Between 2010–2013, 1538 patients with pT2 (Fuhrman grade (FG) 3–4) or pT3-T4 or N+ disease were randomized to pazopanib 800 mg daily or placebo for one year. Again, due to toxicity, this was later amended to a lower starting dose of 600 mg daily because of the superior quality-of-life and safety results of the COMPARZ trial conducted in metastatic ccRCC [25]. With a median follow-up of just over 30 months in each arm, the study did not meet its primary DFS end point in the 600 mg pazopanib vs. placebo group (HR, 0.86 (95% CI 0.70–1.06, *p* = 0.16)). Of note, among patients who received 800 mg pazopanib (N = 198) there was an improvement in DFS (HR, 0.69 (95% CI 0.51–0.94, *p* = 0.02)) [24]. However, there was no improvement in OS in either the 600 mg or 800 mg pazopanib group vs. placebo (HR, 1.0 (95% CI 0.80–1.26, *p* > 0.9)) [26]. Remaining disease free at 2 years was prognostic for OS (5-year OS rate, (95% CI) 0.51 (0.45–0.57), *p* < 0.0001). Discontinuation rates were similar in both the 600 mg and 800 mg treatment groups (35% and 39%, respectively). The most common AEs for discontinuation were elevated liver enzymes (ALT and AST), and grade 3 to 4 AEs occurred in 60% of patients in the pazopanib 600 mg group compared to only 21% in the placebo group with hypertension (52% vs. 19%), diarrhea (64% vs. 25%) and AST elevation (35% vs. 5%) being the most common [24].

### 3.3. Axitinib

The ATLAS (adjuvant axitinib therapy of renal cell cancer in high-risk patients) international phase III trial evaluated axitinib vs. placebo in patients with locoregional predominant ccRCC after CRN [27]. Between 2012–2016, 724 patients with any grade ≥ pT2 and/or N+ disease were randomized (axitinib 5 mg twice daily, dose adjustments were permitted) or placebo for 1–3 years. The primary objective was DFS according to independent review committee (IRC) assessment; furthermore, the trial included a prespecified subgroup analysis of DFS in the highest (pT3 with FG ≥ 3 or pT4 and/or N+ any FG) or lowest (pT2 or pT3 with FG ≤ 2) risk subpopulations [27]. There was no improvement in DFS by IRC on interim analysis (HR, 0.87 (95% CI 0.620–1.66, *p* = 0.9483)) and the trial was stopped early. In the prespecified subgroup analysis, the IRC assessment also found no reduction in risk of a DFS event in the lower-risk subpopulation (HR, 1.016 (95% CI 0.620–1.66, *p* = 0.9483)), but per investigator assessment, in higher-risk disease there was a benefit (HR, 0.641 (95% CI 0.468–0.879, *p* = 0.0051)) [27]. Treatment discontinuations took place in 57% of patients in the axitinib-treated group and 49% in the placebo-treated group; 56% of patients in the axitinib-treated group experienced an AE. Grade 3 or 4 AEs were reported in 49% of patients in the axitinib-treated group with hypertension (16%), diarrhea (11%), fatigue (11%), and palmar–plantar erythrodysesthesia (5%) being the most frequent [27].

### 3.4. VEGFR-TKI Trials in the M1 Setting

Two clinical trials have explored the application of adjuvant VEGFR TKI after resection of metastatic sites before or after CRN. The RESORT phase II trial also evaluated the role of adjuvant sorafenib vs. observation, though in the setting of advanced ccRCC after total metastasectomy (three or fewer lesions) and CRN [28]. A total of 76 patients were randomized (between 2012–2017) within 12 weeks of a metastasectomy to sorafenib 400 mg twice daily or observation for a maximum of 52 weeks. With a median follow-up of 38 months, there was no improvement in RFS with sorafenib vs. placebo (95% CI 20-NA, *p* = 0.404). AEs were higher in the sorafenib arm vs. observation (84% vs. 31%, respectively), with a treatment discontinuation rate of 19% in the sorafenib arm. With 42 months follow-up, median RFS was 21 months in the sorafenib arm and RFS probability was 32% (95% CI 18–57%) at 48 months (HR, 1.35 (95% CI 0.72–2.54)) compared to the observation arm where median RFS was 35 months and RFS probability was 44% (95% CI 30–65%) at 48 months [29].

Pazopanib was also studied in the advanced ccRCC setting in patients after CRN who had undergone total metastasectomy in the phase III ECOG study E2810 [30]. A total of 129 patients were randomized (between 2012–2017) to receive pazopanib 800 mg daily vs. placebo for one year. There was no limit to the number of resected metastatic sites, but successful removal had to be 2–12 weeks prior to randomization and any surgery (including CRN) had to have been completed 12 weeks prior to registration. Median follow-up from time of randomization was 30 months (range 0.4–66.5 months). Pazopanib treatment for 1 year after total metastasectomy did not improve DFS (HR, 0.85 (95% CI 0.55–1.31, *p* = 0.47)) or OS (HR, 2.65 (95% CI 1.02–6.9, *p* = 0.05)) [30].

### 3.5. Adjuvant mTOR Treatment

The EVEREST (everolimus for renal cancer ensuing surgical therapy) phase III Southwest Oncology Group (SWOG) intergroup trial evaluated everolimus vs. placebo in patients with clear cell or non-clear cell RCC after CRN [6]. Between 2011 and 2015, 1464 advanced RCC patients with intermediate/high (pT1 FG 3–4 N0 to pT3a FG 1–2 N0) or very high risk (pT3a FG 3–4 to pT3 FG any or N+) received 10 mg everolimus daily or placebo for 1 year. The sample size was originally planned to be 1170 patients; however, an unequal number of discontinuations in the everolimus arm warranted an increase to the final enrollment of 1464. The primary endpoint was RFS for the total population and included two prespecified subgroup analyses in the very high-risk and intermediate high-risk subpopulations. With a median follow of 76 months, RFS trended toward significance in the everolimus group vs. placebo (HR, 0.85 (95% CI 0.72–1.00, *p* = 0.0246)) with a one-sided significance level of 0.022. Of note, RFS was improved with everolimus vs. placebo in the very high-risk subpopulation, pT3a G3-4 or pT4 G any or N+, (HR, 0.79 (95% CI 0.65–0.97, *p* = 0.011)), but no signal was seen in the intermediate/high-risk subpopulation (HR, 0.99 (95% CI 0.73–1.35, *p* = 0.48)) [6]. AEs were greater in the everolimus group with 46% of patients reporting a grade 3 or 4 AE, most commonly mucositis (14%), hypertriglyceridemia (11%), and hyperglycemia (5%).

## 4. Immune Checkpoint Inhibitor Adjuvant Clinical Trials

ICIs are humanized monoclonal antibodies that bind to key immune checkpoint proteins including programmed death protein-1 (PD-1) or its ligand (PD-L1), and cytotoxic T-lymphocyte antigen-4 (CTLA-4). Blocking these immune-suppressive proteins boosts immune recognition and effective anti-tumor immunity. This phenomenon of continuous immune recognition even after treatment discontinuation made ICIs attractive in the ccRCC adjuvant setting. Current clinical trial data regarding the use of ICIs for intermediate or high-risk ccRCC after CRN have presented conflicting results, and descriptions of these trials are described below with pertinent findings summarized in Table 2.

### 4.1. Nivolumab

Nivolumab is a humanized monoclonal antibody that targets the PD-1 protein and is approved as a single agent in refractory metastatic RCC following previous VEGF-targeted therapy based on results from the phase III CheckMate-025 trial, which showed an improvement in OS for nivolumab compared to everolimus (25 vs. 19.6 months, (HR, 0.73 (98.5% CI 0.57–0.93, *p* = 0.0018))). Nivolumab therapy was well-tolerated with the most frequent AEs reported as fatigue (33%), nausea (14%), and pruritus (14%) [31]. Health-related quality of life (HRQoL) was improved with nivolumab vs. everolimus by the Functional Assessment of Cancer Therapy–Kidney Symptom Index–Disease-Related Symptoms score (FKSI-DRS). The rate of improvement was 55% by FKSI-DRS for nivolumab vs. 37% for everolimus [32]. Due to these encouraging results in treating metastatic RCC, two large trials have now studied nivolumab in the perioperative setting before and after RN.

The role of neoadjuvant or perioperative nivolumab has been studied in a phase I clinical trial at Johns Hopkins [33]. The investigators enrolled 17 patients with biopsy-confirmed high-risk ccRCC (T2a-T4NanyM0 or TanyN1M0) prior to nephrectomy to receive three doses of neoadjuvant nivolumab. Ten (58.8%) patients experienced an AE (all grade 1–2) potentially attributable to nivolumab treatment. All evaluable patients had stable disease per radiographic criteria, with one demonstrating an immune-related pathologic response. Metastasis-free survival and OS were 85.1% and 100% at 2 years, respectively [33]. Given these results, the PROSPER (phase III randomized study comparing perioperative nivolumab vs. observation in patients with RCC undergoing nephrectomy) ECOG intergroup trial E8143 evaluated perioperative and adjuvant nivolumab vs. observation in high-risk RCC [10]. Between 2017–2021, 819 RCC patients (clinical stage ≥ T2 or Tany N+ or M1NED within 12 weeks of nephrectomy) were randomized to surgery alone followed by observation, or nivolumab (480 mg IV) for one dose followed by surgery (7–28 days later) and then adjuvant nivolumab every 4 weeks for up to 9 doses. Confirmatory renal or metastasectomy biopsies were required only from patients assigned to the nivolumab arm if no biopsy confirmation was performed 12 months prior to randomization. The primary endpoint was RFS regardless of RCC histological subtype, with secondary endpoints being OS, QoL, and ccRCC-specific RFS. Those patients who did not have surgery or still had disease present following surgery were considered an event at day 1. Of the enrolled patients, 53% had cT2 disease with 68% having >cT3 and 83% had ccRCC. Of 404 patients assigned to nivolumab, only 353 pursued nivolumab and 359 received surgery, while in the observation arm only 387 of 415 patients went on to surgery. At time of surgery, 77% had ccRCC while 32% had pT2 or less, and 3% of patients who had surgery were not disease-free following surgery. The trial ended early because of futility and RFS was not improved with nivolumab (HR, 0.97 (95% CI 0.74–1.28, *p* = 0.43)) [10]. OS was not mature at the time of the study evaluation. Reported AEs attributed to nivolumab occurred in 20% of patients with grade 3–4 AEs including rash, kidney injury, and elevated lipase. Multiple correlative studies are planned to include absolute T-cell clonotype abundance, changes in T-cell repertoire, T-cell receptor diversity, and changes in cytokine levels for patients randomized to nivolumab. These correlatives may help identify potential biomarkers to predict response and may correlate with RFS and OS.

Given the activity of nivolumab with ipilimumab in the metastatic setting, with an ORR of 40% and median OS of 47 months in those with intermediate or poor risk ccRCC, the CheckMate-914 international 2-part phase III trial evaluated this combination (NIVO + IPI) vs. placebo (Part A), or nivolumab monotherapy vs. NIVO + IPI vs. placebo (part B) in patients with high-risk localized RCC after CRN [8]. To date, only the Part A trial results have been reported. A total of 816 patients were randomized in Part A with predominant ccRCC with pathologic stage T2a (FG 3 or 4), >T2b (FG any), or any T (FH any) N1. Patients received nivolumab every 2 weeks (12 doses) plus ipilimumab (1 mg/kg) every 6 weeks (4 doses) or placebo for 24 weeks after surgery. The primary endpoint was DFS per blinded independent review and secondary endpoints were OS and safety. With a median follow-up of 37 months, there was no improvement in DFS for NIVO + IPI vs. placebo (HR, 0.92 (95% CI 0.71–1.19, *p* = 0.5347)) [8]. There were notable toxicities: AEs were reported in 88.9% (≥grade 3 in 28.5%) of patients who received NIVO + IPI vs. 56.8% (≥grade 3.2%) of patients receiving placebo. Treatment-related AEs that led to early discontinuation of NIVO + IPI occurred in 33% of patients compared to only 1% of patients receiving placebo, and only 57% of patients completed 6 months of therapy [8]. Efficacy results for patients enrolled in Part B are still ongoing.

### 4.2. Pembrolizumab

Pembrolizumab is an anti-PD-1 monoclonal antibody and it is approved in combination with axitinib or lenvatinib based on the KEYNOTE-426 and CLEAR trials, respectively, for first-line treatment in the advanced setting and as monotherapy in the adjuvant setting in ccRCC [9,34,35]. The KEYNOTE-564 international phase III trial compared adjuvant pembrolizumab to placebo in ccRCC patients at high risk for recurrence (T2 FG 4, >T3 or N+, M1 NED at time of nephrectomy or within 1 year) after CRN. A total of 1406 patients were randomized between 2017–2019 to receive 200 mg pembrolizumab or placebo every 3 weeks for 1 year (maximum 17 cycles) [9]. The primary endpoint was DFS per investigator’s assessment, with OS and safety as secondary endpoints. Median follow-up from time to randomization was 24.1 months. On first interim analysis, there was a longer DFS with pembrolizumab therapy vs. placebo (HR, 0.68 (95% CI 0.53–0.87, *p* = 0.002)) [9]. Subsequent analysis performed at the 30-month follow-up still demonstrated improved DFS with pembrolizumab vs. placebo (HR, 0.63 (95% CI 0.50–0.80)) with median DFS not reached in either group [36]. OS results are immature at this time (only 33% of death events have been reported) and additional follow-up is needed, though from 24 to 30 months of follow-up more events occurred in the placebo arm than in the experimental arm. At extended follow-up, the estimated proportion of enrolled patients alive without disease progression on subsequent treatment was 92.5% (95% CI 89.7–94.5) in the pembrolizumab group vs. 86.1% (95% CI 82.5–89.1) in the placebo group [9]. Patients tolerated adjuvant pembrolizumab well with no new safety signals reported. Grade 3 or higher AEs of any cause were reported in 32% of patients receiving pembrolizumab vs. 18% of patients receiving placebo. Immune-related adverse events (irAEs) happened in 36% of patients receiving pembrolizumab and 7% receiving placebo, with the most common being abnormal thyroid function. Only 8% of the patients receiving pembrolizumab required high-dose steroids (defined as >40 mg prednisone equivalent daily) vs. 1% receiving placebo [9,36].

### 4.3. Atezolizumab

Atezolizumab is an anti-PD-L1 monoclonal antibody and has been studied in the adjuvant and metastatic settings in multiple solid tumors, including RCC. The IMmotion010 international phase III trial compared adjuvant atezolizumab vs. placebo in intermediate or high-risk patients (T2 FG 4, T3 FG 3 or 4, >T3b FG any, Tany N+ and M1 NED) with a clear cell component or sarcomatoid histology after CRN [7]. In contrast to KEYNOTE-564, IMmotion010 enrolled a different M1 population that included both synchronous metastasectomy and metachronous metastasectomy occurring over 12 months. The trial enrolled 778 patients between 2017–2019 to receive 1 year of atezolizumab (1200 mg IV every 3 weeks) or placebo. The primary endpoint was investigator-assessed DFS. Median follow-up from time of registration was 44.7 months. There was no improvement in DFS with atezolizumab vs. placebo (HR, 0.93 (95% CI 0.75–1.15, *p* = 0.50)). No improvement was seen in OS on initial analysis. AEs led to treatment discontinuation in 12% of patients in the atezolizumab group and 3% in the placebo group. irAEs were reported in 54% and 28% of patients receiving atezolizumab and placebo, respectively [7]. The most common irAEs that happened in 15% of patients receiving atezolizumab were rash and hypothyroidism.

## 5. Patient Selection for Adjuvant Therapy

Accurately predicting early disease progression is important so that physicians can select the patients most likely to benefit from adjuvant therapies. Several existing risk models rely not only on the tumor-node-metastases (TNM) stage, but also incorporate ECOG performance status and the Fuhrman grade. Currently, four such models available for use in localized RCC, namely the UISS, MSK post-operative nomogram, SSIGN score, and Leibovich score; the details are listed in Table 3 [22,37,38,39]. These models help to stratify patients into risk groups (low, intermediate, or high) and have been incorporated into eligibility criteria for past adjuvant trials. Unfortunately, these models are broad, rely on retrospective or historical data spanning decades, and are not personalized to the individual patient, with rates of relapse varying greatly even within each risk group. A modern approach to consider for risk stratification is the ASSURE RCC prognostic nomogram [40]. This model generates three predictions (early disease progression, DFS, and OS) for kidney cancer patients after nephrectomy. In addition to the TNM stage, it also incorporates histology, presence or absence of vascular invasion, age, coagulative necrosis, and sarcomatoid features [40]. These models can be utilized during adjuvant therapy discussions and complimentary approaches are being explored. Lastly, recent data has emerged regarding grouping RCC tumors into seven transcriptional and genomic alteration (GA) clusters: angiogenic/stromal, angiogenic, complement/fatty oxidation, T-effector/proliferative, proliferative, stromal proliferative, and snoRNA [41]. These clusters were combined with previously reported data categorizing RCC tumors into subgroups based on relative expression of angiogenesis, immune subtypes, and myeloid inflammation-associated genes [42]. These first subgroups or signatures were described as potential molecular correlates of response in advanced RCC from the phase II IMmotion150 trial, which explored atezolizumab with or without bevacizumab vs. sunitnib [42]. These clusters were then applied to RCC tumors from the phase III IMmotion151 trial and have been associated with clinical outcomes depending on treatment (anti-VEGF or anti-PD-L1). The authors were able to conclude that clusters in the T-effector group achieved clinical benefit from the combination of atezolizumab and bevacizumab. In addition, sarcomatoid tumors demonstrated lower incidence of *PBRM1* mutations, angiogenesis markers, *CDKN2A/B* alterations, and PD-L1 expression, which helps to develop possible molecular subtypes for further RCC tumor characterization [41]. These clusters, while intriguing, do need prospective validation in the adjuvant RCC setting.

Ultimately, clinical prognostic models are imperfect, and minimally invasive laboratory methods are needed to classify patients at risk of metastatic relapse. So-called ‘circulating biomarkers’ are currently being investigated in RCC patients to identify those at highest risk of recurrence. Kidney injury molecule-1 (KIM-1) is a transmembrane glycoprotein expressed by injured renal proximal tubular cells and RCC cells that can be measured in a patient’s urine or plasma [43,44]. Studies have found that KIM-1 levels can also be present in an RCC patient’s blood up to 5 years after CRN and may be associated with survival [45]. Subsequently, investigators performed a retrospective analysis of ASSURE patients (with blood available following a total CRN) and reported an association with higher baseline plasma detection of KIM-1 and worse OS in a multivariable accelerated failure time (AFT) model, after adjustment for age, sex, performance status, Fuhrman grade, nodal stage, tumor stage, presence of sarcomatoid features, and tumor histology (survival time ratio 0.71 for 75th vs. 25th percentile of KIM-1; 95% CI, 0.56–0.91; *p* < 0.001) [46]. KIM-1 levels are also being evaluated retrospectively in patients enrolled in the PROSPER trial.

## 6. Circulating Tumor DNA Testing in RCC

Circulating tumor DNA (ctDNA) has emerged as a biomarker for adjuvant therapy benefit in urothelial carcinoma and is being studied in RCC patients [47]. While ctDNA from circulating cancer cells can be found in the plasma of cancer patients [48,49], several studies have found that ctDNA levels in RCC are typically lower in plasma compared to other genitourinary tumors [49]. In the metastatic RCC setting, detection of GAs in the blood through ctDNA was seen in anywhere from 28–78% of patients and a wide range of ctDNA detection was corroborated in a meta-analysis examining 19 studies [50,51,52]. Other studies have attempted to correlate GAs in plasma utilizing ctDNA next-generative sequencing panels. These studies found that only 30% of RCC patients had detectable ctDNA [53,54]. However, Yamamoto et al. found that in metastatic RCC patients positive ctDNA, short fragment size of ctDNA, and a high proportion of cell free fragments were factors significantly associated with worse cancer-specific survival. These findings were not significant in RCC patients without metastases, necessitating further investigation into how to incorporate ctDNA into the adjuvant setting [53]. Furthermore, results utilizing bespoke analyses from the perioperative setting have been disappointing. In an analysis of 34 patients undergoing CRN, 41% (14/34 patients) had detectable ctDNA prior to nephrectomy. Positive ctDNA was associated with increased tumor size, stage, and poorly differentiated tumors [55]. Post-operative samples were available in 41 patients, with a median follow-up of 64 months. The positive predictive value of having ctDNA positivity was 100% (8 of 8 patients positive for ctDNA relapsed); the negative predictive value was only 52% (16/33 ctDNA-negative patients relapsed). Furthermore, 17 patients who did not have disease relapse were ctDNA negative (100% specificity) and among 24 with relapse, only 8 were ctDNA positive (33% sensitivity), highlighting the need for more sensitive biomarkers [55]. To that end, cell-free methylated-DNA immunoprecipitation and high-throughput sequencing (cfMeDIP–seq) is proving to be a highly sensitive assay for early state RCC tumor detection. Data from Nuzzo et al. have shown this method can accurately classify patients across all stages of RCC and demonstrated the validity of this assay to identify patients with RCC using urinary cell-free DNA [56]. This method will need to be validated in a prospective fashion, but the early results are encouraging.

## 7. Discussion

The advances made in treatment for metastatic RCC have failed to translate into a definitive benefit in the adjuvant setting. Without an OS benefit in the STRAC trial and with neither DFS nor OS in the other five phase 3 trials of adjuvant VEGFR TKIs, the US National Comprehensive Cancer Network (NCCN) kidney cancer guidelines label sunitinib a Category 3 recommendation and its adjuvant use has fallen out of favor in clinical practice [17]. This recommendation is further supported by two meta-analyses that showed adjuvant TKI monotherapy not only provides no DFS (HR, 0.92 (95% CI 0.86–1.00)) or OS benefit (HR, 1.01 (95% CI 0.91–1.12)) but considerably increases toxicity [57,58]. Furthermore, despite intriguing results from the EVEREST trial, everolimus is unlikely to be adopted in the adjuvant RCC setting given no statistically significant improvement in DFS or OS with considerable toxicity. Table 4 compares the two FDA-approved treatments in adjuvant high-risk ccRCC, highlighting key points from each completed clinical trial.

Adjuvant ICI therapy is more promising, but recent results have clouded their future. To date, only pembrolizumab has been approved in the adjuvant setting in intermediate or high-risk ccRCC, while three other trials recently failed to meet their primary endpoints: perioperative nivolumab for one dose followed by 9 months of nivolumab (PROSPER), 6 months of adjuvant NIVO + IPI (CHECKMATE-914), or 12 months of adjuvant atezolizumab (IMMotion-010) did not improve DFS, while OS data are immature for all trials. Ultimately, further analysis of the data is imperative. The activity of these agents in the metastatic setting varies, with responses for pembrolizumab exceeding 36%, but nivolumab and atezolizumab monotherapy were both below 30% [31,59,60]. Each trial also had varying degrees of design differences that may account for their conflicting results compared to KEYNOTE-564. For example, PROSPER used a perioperative strategy to select patients based on clinical, not pathologic, T-stage and allowed variant histologies. Checkmate-914 explored a shorter course with nivolumab given at traditional doses, but ipilimumab was administered every 12 weeks and excluded all M1 NED patients. IMmotion010 utilized a PD-L1 inhibitor and enrolled patients with oligometastatic disease more than 12 months after primary nephrectomy. As such, longer follow-up and analysis of all data will be critical.

Additionally, subgroup and correlative analysis could potentially identify if predictive biomarkers can be established. Data from KEYNOTE-564 shows a benefit across all RCC subgroups and similar analysis of other trials will be imperative, as will planned correlative studies [36]. In addition to biomarkers of ctDNA to determine which patients are at highest risk for recurrence, correlative analysis looking at tumor and host markers of response will be important to select the right therapeutic modality. Except for nivolumab + ipilimumab, each of these studies demonstrated that ICIs in the adjuvant setting are tolerable and most patients can complete the treatment period (compared to VEGFR TKIs), and no new safety signals were established. To date, there has not been an improvement in OS with any of the trials, though there is a trend for adjuvant pembrolizumab. As we await OS results, in a recent retrospective analysis of the SEER database Kendall’s τ rank showed correlation between DFS and OS (Kendall’s τ = 0.70; 95% CI: 0.65–0.74; *p* < 0.001), suggesting that DFS is a reasonable surrogate endpoint while OS is immature [61].

Ultimately, the recent reporting of these ICI adjuvant trials may impact clinical and patient decision-making regarding the utilization of adjuvant pembrolizumab. Patients may be less likely to consider adjuvant treatment because of these current findings, but a discussion is still merited for patients who meet the KEYNOTE-564 eligibility criteria (T2 G4, T3–T4 any grade, Any TxN1, M1NED within 1 year). Validated prognostic biomarkers are not widely available in clinical practice outside of clinical trials, but several promising models are being explored as detailed above. In the future, we anticipate that the combination of biomarkers, when validated, will be imperative to help guide patient selection outside enrollment of ongoing clinical trials. Toxicities must be discussed and reviewed with patients prior to the start of adjuvant immunotherapy, in addition to the benefits of therapy. Active surveillance still plays a role for some patients after CRN if criteria are not met for adjuvant treatment as outlined above.

In addition, efforts are being made to build upon the success of pembrolizumab in the adjuvant setting. Table 5 provides a list of ongoing trials investigating ICIs and other agents in combination in the neoadjuvant or adjuvant settings. RAMPART is an international multi-arm multi-stage platform trial with a co-primary endpoint of OS and DFS investigating PD-L1 and CTLA4 inhibitors for clear cell and non-clear cell RCC patients in the adjuvant setting [62]. LITESPARK-022 is currently enrolling patients with RCC after CRN akin to KN-564 criteria, with the exception that patients who are M1 NED can undergo metastasectomy up to 2 years after nephrectomy, and it randomizes patients to the HIF-2α inhibitor, belzutifan plus pembrolizumab vs. placebo plus pembrolizumab. Additional trials are in development exploring other combinations in this space [63].

## 8. Conclusions

Adjuvant therapy in patients with high-risk ccRCC is evolving. While, to date, no therapeutic intervention has shown an improvement in OS, both a VEGFR TKI and an ICI are FDA approved in this space; however, given VEGFR TKI toxicities, pembrolizumab is the preferred option at this time per NCCN guidelines. Subsequently, informed discussions between oncologists and patients are imperative and the decision to pursue adjuvant therapy or to defer must be a shared decision. Patients with high-risk ccRCC should be encouraged to enroll in ongoing adjuvant or neoadjuvant ICI clinical trials and further investigation of competed clinical trials to find clinical and biologic markers of response will be imperative to build upon the advances made to date.

## Figures and Tables

**Table 1 cancers-14-06018-t001:** Completed Adjuvant Phase 3 Trials with VEGFR TKIs in RCC.

Trial Name (Year)	ASSURE (2016)	S-TRAC (2016)	PROTECT (2017)	ATLAS (2017)	SORCE (2020)
Agent Investigated	Sunitinib ^ǂ^ vs. Sorafenib ^ǂ^ vs. Placebo	Sunitinib ^ǂ^ vs. Placebo	Pazopanib ^ǂ^ vs. Placebo	Axitinib * vs. Placebo	Sor 3yr vs. Sor 2yr vs. Placebo (3/2yr)
DFS HR (95% CI, *p*-value)	1.02 (0.85–1.23, *p* = 0.80)	0.76 (0.59–0.98, *p* = 0.03)	0.86 (0.70–1.14, *p* = 0.17)	0.87 (0.66–1.15, *p* = 0.32)	1.01 (0.83–1.23, *p* = 0.95)
OS HR (95% CI, *p*-value)	Sun: 1.06 (0.78–1.44) Sor: 0.80 (0.58–1.10)	0.92 (0.66–1.28; *p* = 0.6)	1.0 (0.80–1.26, *p* > 0.9)	1.026 (0.60–1.756, *p* = 0.9246)	Sor 3yr 1.06 (0.82–1.38, *p* = 0.638) Sor 1yr 0.92 (0.71–1.20, *p* = 0.541)
Treatment related AEs Grade 3–5 (%)	Sun: 63% Sor: 72% Pbo: 25%	Sun: 60.4% Pbo: 19.4%	Paz: 60% Pbo: 21%	Axi: 49% Pbo: 12%	Sor 3yr: 63.9% Sor 1yr: 58.6% Pbo: 29.2%
Treatment discontinuation (%)	Sun: 44% Sor: 45% Pbo: 11%	Sun: 44% Pbo: 31%	Paz 600mg: 35% Paz 800mg: 39% Pbo: 5/6%	Axi 1yr: 67.9% Pbo 1yr: 72.4% Axi 3yr: 79.8% Pbo 3yr: 78.8%	Sor 3yr: 75% Sor 1yr: 67% Pbo: 45%

^ǂ^ = 1 year treatment; * = up to 3 years; AEs = Adverse Events; Sor = Sorafenib; Sun = Sunitinib; Paz = Pazopanib; Axi = Axitinib; Pbo = Placebo.

**Table 2 cancers-14-06018-t002:** Completed Adjuvant Phase 3 Trials with Immune Checkpoint Inhibitors in RCC.

Trial Name (Year)	IMmotion010 (2022)	Checkmate-914 * (2022)	PROSPER (2022)	KEYNOTE-564 (2019)
Investigational Agents	Atezolizumab vs. Placebo	Nivolumab + Ipilimumab vs. Placebo (Part A)	Perioperative + adjuvant nivolumab vs. observation	Pembrolizumab vs. Placebo
RCC Inclusion Histology	Clear cell, sarcomatoid	Clear cell	Any	Clear Cell
M1 NED population included (%)	14.4%	0	3%	5.8%
DFS HR (95% CI, *p*-value)	0.93 (0.75–1.16, *p* = 0.50)	0.92 (0.71–1.19, *p* = 0.5347)	0.97 (0.74–1.28, *p* = 0.43)	0.63 (0.50–0.80, *p* = n/a)
Treatment related AEs, Grade 3–4 (%)	14.1%	28%	15%	18.6%
Treatment discontinuation (%)	11.5%	29%	13%	18.2%

NED = no evidence of disease; AEs = Adverse Events; * = Part B results are not reported.

**Table 3 cancers-14-06018-t003:** RCC Risk Models for Patient Selection after cytoreductive nephrectomy.

UISS Model [37]	TNM Stage	--	Fuhrman Grade	--	--	ECOG PS	1–5 Year OS%
MSK postoperative nomogram [38]	Pathologic stage	Tumor size	Fuhrman grade	Vascular invasion	Clinical presentation	--	5 year RFS%
SSIGN score [39].	TNM stage	Tumor size	Nuclear grade	Tumor Necrosis	--	--	5 year CSS %
Leibovich score [22]	TNM stage	Tumor size	Nuclear grade	Tumor Necrosis	--	--	1,5,10 year MFS %

OS = Overall Survival; CSS = Cancer Specific Survival; MFS = Metastases Free Survival; TNM = Tumor Node Metastases.

**Table 4 cancers-14-06018-t004:** Comparison of approved adjuvant treatments after radical nephrectomy for high-risk RCC.

Trial Name (Enrollment Periods)	S-TRAC [5] (9/19/07-4/7/11)	KEYNOTE-564 [9] (6/30/17-9/20/19)
Investigational Agents	Sunitinib vs. Placebo	Pembrolizumab vs. Placebo
Enrolled Patients	615	994
Median Follow Up (Months)	80	30
Median DFS-Investigator Review-HR (95% CI, p-value)	0.81 (0.64–1.02, *p* = 0.08)	0.63 (0.5–0.80, *p* <0.0001)
Median DFS (Central review) HR (95% CI, *p*-value)	0.76 (0.59–0.98, *p* = 0.03)	NR
24-month DFS Rate (%)	72% vs. 68%	78.3% vs. 67.3%
Median OS HR (95% CI, *p*-value)	NR vs. NR 0.92 (0.66–1.28, *p* = 0.60)	NR vs. NR * 0.52 (0.31–0.86, *p* = 0.0048)
Grade 3 or 4 TRAEs (%)	57.2	18.9
Treatment Discontinuation due to AEs (%)	28	20.7

NA = not available; NR = not reached; DFS = disease free survival; OS = overall survival; TRAE = treatment related adverse events; AEs = adverse events; * = Did not cross prespecified p-value boundary for statisticial signifiance for 51 events. Final overall survival will be reported after 200 events.

**Table 5 cancers-14-06018-t005:** Enrolling Neoadjuvant or Adjuvant Clinical Trials in High Risk RCC.

Trial Name (Sponsor)	NCT#	Investigational Agents	Planned Accrual	Inclusion Stage/Grade	Histology	Primary Endpoint
LITESPARK 6482-022 (MERCK)	05239728	Belzutifan + pembrolizumab vs. Placebo + pembrolizumab	1600	pT2, FG4 or scarcomatoid, >pT3, FG any N+, and M1 NED	Clear Cell (sarcomatoid allowed)	DFS (investigator)
RAMPART (University College London)	03288532	Durvalumab + tremelimumab vs. Durvalumab vs. Active survillence	1750	Leibovoch Score 3–11	Any	DFS and OS

## Data Availability

Not applicable.

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
