# Peer review of "The Continuing Question of Adjuvant Therapy in Clear Cell Renal Cell Carcinoma"

_cancers, 2022, doi:10.3390/cancers14246018_

Round 1

Reviewer 1 Report

The manuscript draft contains brief account of current therapies and ongoing clinical trials for RCC. The review is very well written with attention to detail and potential impact of adjuvant therapy in clear cell renal cell carcinoma. 

The manuscript can be accepted in its present form. Nonetheless, it would be great if authors could add their view on future studies and research breakthroughs that would impact the field at the end of the conclusion (For example the ongoing discoveries of biomarkers as mentioned in the manuscript). I do understand the focus of the manuscript is the clinical trials but the lesson learned from these and a way forward would be much appreciated by the readers.  

Reviewer 2 Report

Well-written comprehensive review of adjuvant treatment options in clear cell renal cell carcinoma. Recommend accepting for publication in current form. 
